# Deep Binary Classification via Multi-Resolution Network and Stochastic Orthogonality for Subcompact Vehicle Recognition

**DOI:** 10.3390/s20092715

**Published:** 2020-05-09

**Authors:** Joongchol Shin, Bonseok Koo, Yeongbin Kim, Joonki Paik

**Affiliations:** Department of Image, Chung-Ang University, Seoul 06974, Korea; mbstel275@gmail.com (J.S.); izg2sd@gmail.com (B.K.); sawors2010@gmail.com (Y.K.)

**Keywords:** vehicle recognition, multi resolution network, optimization

## Abstract

To encourage people to save energy, subcompact cars have several benefits of discount on parking or toll road charge. However, manual classification of the subcompact car is highly labor intensive. To solve this problem, automatic vehicle classification systems are good candidates. Since a general pattern-based classification technique can not successfully recognize the ambiguous features of a vehicle, we present a new multi-resolution convolutional neural network (CNN) and stochastic orthogonal learning method to train the network. We first extract the region of a bonnet in the vehicle image. Next, both extracted and input image are engaged to low and high resolution layers in the CNN model. The proposed network is then optimized based on stochastic orthogonality. We also built a novel subcompact vehicle dataset that will be open for a public use. Experimental results show that the proposed model outperforms state-of-the-art approaches in term of accuracy, which means that the proposed method can efficiently classify the ambiguous features between subcompact and non-subcompact vehicles.

## 1. Introduction

Typically, subcompact cars are defined by the engine displacement, width, and height under 1000 cc, 1.6 m, and 2.0 m, respectively. To satisfy these specifications, the subcompact car has a unique shape such as shorter-bonnet and hatchback. In addition, there are various environmental benefits because the subcompact cars have a small displacement engine and a light weight. To encourage people to drive subcompact cars, many countries provide several benefits—discounts on tall road charge and parking fee. Since classification of subcompact cars from other requires labor-intensive human investigation, an automatic vehicle classification system is needed. In general, vehicle classification methods can be classified into two approaches: one uses infrared sensors to measure physical dimensions of a vehicle such as length, height, and width. The other uses a single camera and image processing algorithms to recognize the visual characteristics of vehicles [1,2]. Despite of accuracy and robustness, the infrared sensor-based system is too expensive to be installed in many places. Thus, we propose an image recognition system to reduce the installation and maintenance cost. To classify the visual feature in images, Dalal et al. extracted the histogram of oriented gradients (HOG) and classify the HOG using support vector machine (SVM) [3]. To the best of authors’ knowledge, the HOG-based SVM is the most popular approach to recognize objects before deep learning has become popular. To enhance HOG features that are affected by rotation, or distance, and occlusion, various approaches were proposed. Llorca et al. proposed vehicle manufacturer recognition by detecting the vehicle-logo, but a subcompact car can not be completely classified using only manufacturer information [4]. Clady et al. recognized the vehicle type by separating objects and the background in interactively selected regions [5]. This method is robust to the variance in the distance. However, the region should be passively selected. Mohottala et al. created vehicle images using computer graphics (CG), and then classify the type of vehicle using eigenvalues [6].

Although this approach can easily obtain the vehicle data, it cannot avoid error in real vehicle data. Michael and Daniel classified the eigenvalues of vehicle classes using neural networks [7]. Since they used an artificial neural network, classification accuracy was acceptable only without occlusion. Huttunen et al. adaptively recognized the vehicle classes using a deep neural network [8]. This method can recognize the multi-class vehicles such as sedan, truck, and bus. However, in subcompact car classification, it has overffiting while learning the subcompact vehicle class because it is difficult to discriminate the subcompact vehicle from others. Simonyan et al. proposed the deep convolutional neural networks called VGG16 and VGG19 [9]. Since the VGG networks can be pre-trained via a large-scale image dataset [10], it can have a very deep hidden-layer to recognize the vehicle. However, it cannot robustly classify the subcompact vehicles because of both obscure features and environmental variables as shown in Figure 1. He et al. proposed the more deep residual networks [11]. This network can be designed more deeply such as 50, 101, 151 layers because of the residual learning. However, in the binary classification, the VGG networks are also deep enough. Xie et al. applied the split-transform-merge strategy to deep residual networks [12]. This strategy can effectively recognize various features, but it can not adaptively crop the image region. Karpathy et al. proposed the multiple convolutional neural networks with center clipping and image fusion for video classification [13]. It can recognize the obscure objects and actions in video, but it cannot localize objects that are not in the center of the image.

To solve this isolated problem, we proposed a novel multi-resolution network and stochastic orthogonal learning method. More specifically, the proposed method include three functional steps: (i) we emphasize the features using retinex model-based image-enhancement [14], (ii) we track the bonnet region using an optimized correlation filters [15], and (iii) we engage this region and image using the proposed multi-resolution network. In addition, we learned our muti-resolution network by considering stochastic orthogonality of probabilities between subcompact and general vehicles. We also build a subcompact vehicle dataset including 1500 training data and 2000 test-images. Experimental results show that the proposed method outperforms state-of-the-arts approaches in terms of accuracy by over 12.25%. This paper is organized as follows. In Section 2, we describe the related works. The proposed multi-resolution network is presented in Section 3 followed by experimental results in Section 4, and Section 5 concludes this paper.

## 2. Related Works

### 2.1. Support Vector Machine

To classify the features in images, the SVM can be applied by minimizing as following Equation [16]
(1)$$∴\underset{\vec{w},b}{\arg\min}\frac{1}{2}{∥\vec{w}∥}^{2}-\sum_{i=1}\alpha_{i}\left(y_{i}\left(\vec{w}•{\vec{x}}_{i}+b\right)-1\right),$$
where $\vec{w}$ and *b* represent weight and bias of hyper-plane to classify the features, α is an operator to find the support vectors, and *y* denotes the label such as positive or negative. The optimized hyper-plane of the support vector machine works well, but it should be estimated low-dimensional features such as histograms of gradients and scale-invariant features [3,17] to apply imaging systems.

### 2.2. Neural Network

The neural networks can classify the non-linear features because the each node in hidden-layer discriminates the complicated patterns as shown Figure 2. Each node includes the weight, bias, and activate function such as sigmoid, and relu. These parameters can be easily estimated by simple cost function and chain rule as
(2)$$E_{total}\;=\;\sum \frac{1}{2}{\left(y-f(x)\right)}^{2},$$
and
(3)$$\begin{array}{c} \frac{\partial E_{total}}{\partial w_{*}}=\frac{\partial E_{total}}{\partial out_{o1}}*\frac{\partial out_{o1}}{\partial net_{o1}}*\frac{\partial net_{o1}}{\partial w_{*}},\\ \frac{\partial E_{total}}{\partial b_{*}}=\frac{\partial E_{total}}{\partial out_{o1}}*\frac{\partial out_{o1}}{\partial net_{o1}}*\frac{\partial net_{o1}}{\partial b_{*}}, \end{array}$$
where *f* returns the results of neural networks, $w_{*}$ and $b_{*}$ are weight and bias in *-th node. Therefore, each parameter can be estimated as
(4)$$\begin{array}{c} w_{*}\left(t+1\right)=w_{*}\left(t\right)-\frac{\partial E_{total}}{\partial w_{*}},\\ b_{*}\left(t+1\right)=b_{*}\left(t\right)-\frac{\partial E_{total}}{\partial b_{*}}. \end{array}$$

However, the neural net also has limitation to apply large-scale image classification.

### 2.3. Convolutional Neural Network (CNN)

Since an image has various features such as gradients, color, and intensity information, the convolution operators in the hidden layer are effective to extract the image features [9]. Furthermore, these convolution operator can also be optimized by chain-rule. For example, we visualize the extracted image features in convolution layer using CNN feature simulator [18]. Note that the convolution operator can extract the large scale features and textures as shown Figure 3. In other words, the CNN can not only classify the multi-class image but also recognize the detail textures. Therefore, the CNN can be applied to various field using the transfer learning method such as medical imaging [19], intelligent transportation system [20,21], and remote sensing [22]

## 3. Proposed Method

### 3.1. Subcompact Vehicle Dataset

To train and test the proposed network, we collected vehicle images using a digital camera (gray-scale) at a parking gate in Seoul, South Korea. Since the camera was installed under the charge machine, images were captured from an angle viewed from below as shown in Figure 4. Furthermore, we collected vehicle images for 1 year and 6 months to reflect various environmental variables such as day light, back light, dust, and night. The collected images were classified into five types including subcompact sedan, subcompact van, subcompact truck, sedan and sport utility vehicle (SUV), and truck and van according to the design and shape. The dataset was split into training and test sets including 1500 and 2000 images, respectively. The goal of this work is the binary classification (subcompact vehicle or not). To this end, each set of the proposed dataset is divided into subcompact and non-subcompact vehicles as shown in Figure 5.

### 3.2. Pre-Convolution Layer

The proposed network consists of pre-convolution and multi-resolution network layers. In the pre-convolution layer, we resize the original 1920 × 1080 px images to 400 × 300 px, and we amplify the intensity and increase the local-contrast using a simple retinex-based image enhancement algorithm as [14]
(5)$$H\left(x\right)=\frac{I\left(x\right)}{\max\left(l\left(x\right)\varepsilon \right)},$$
where *H* represents high-resolution image, *I* the input gray-image in the dataset, and ε is a very small positive number to avoid division by zero. The illuminance map *l* can be estimated as a smoothing term [23]
(6)$$l=med\left(I\right)-med\left(\left|med\left(I\right)-I\right|\right),$$
where Med represents the local-median filter [24]. Figure 6a,c show that the environmental variables can be normalized, and at the same time local-contrast in the shadow region is also enhanced. To process a low-resolution image, the vehicle region should be localized. In this paper, we detect the region using a correlation filter which has low-computational complexity and efficient localization performance [25]. To reflect a feature of texture, we applied multi-channel correlation filter (MCCF) with the histogram of oriented gradients, which can be defined as ridge regression
(7)$$E\left(w\right)=\frac{1}{2}\overset{N}{\underset{i}{\sum }}\overset{D}{\underset{j}{\sum }}{\left(y_{i}\left(j\right)-\overset{K}{\underset{k=1}{\sum }}w^{\left(k\right)T}x_{i}^{\left(k\right)}\left[\Delta \tau_{j}\right]\right)}^{2}+\frac{\lambda }{2}\sum_{k=1}^{K}{\left(w^{\left(k\right)}\right)}^{2},$$
where $y_{i}\left(j\right)$ is the desired and shifted response in i-th sample $y_{i}={\left[y_{i}\left(1\right),\ldots ,y_{i}\left(D\right)\right]}^{T}$, $x_{i}\left[\Delta \tau_{j}\right]$ is a set of cyclically shifted vehicle images in the training dataset. *N* represents the number of training images, *K* is the channels of feature map including HOG 34-channels, and *w* represents the correlation filter. The response map *y* for a vehicle coordinate in the frequency domain has a Gaussian-shaped distribution centering on a pre-annotated region. Since both input patch and response map are circular matrices for cyclic convolution, the correlation filter *w* can be simply expressed in the frequency domain as
(8)$${\hat{w}}^{*}={\left(\lambda I+\sum_{i}^{N}{{\hat{X}}^{T}}_{i}{\hat{X}}_{i}\right)}^{-1}\sum_{i=1}^{N}{\hat{X}}^{T}{\hat{y}}_{i},$$
where, $\hat{w}$ represents a variable *w* in the frequency domain, * and *T* respectively represent the complex conjugate and transpose of a matrix. The optimized correlation filter can estimate the coordinates of the vehicle region via maximum response-region and distribution as shown in Figure 6d–f. We cropped high-resolution images, based on this picking coordinates to obtain low-resolution images *L* (280 × 200). Note that the proposed pre-convolution layer should be processed in Central processing unit (CPU) for efficient memory allocation.

### 3.3. Multi-Resolution Network

We changed the size of both high- and low-resolution images to 224 × 224 to recognize vehicle type. Note that the scale of the proposed subcompact dataset is gray. Therefore, we generate the 3 zero-min channels with the average color of ImageNet [10], and concatenate 3 zero-min channels to create a pseudo color as
(9)$$H^{R}=H-0.4850,\;H^{G}=H-0.4580,\;H^{B}=H-0.4076,$$
and
(10)$$L^{R}=L-0.4850,\;L^{G}=L-0.4580,\;L^{B}=L-0.4076,$$
where *H* and *L* are single gray-scale (224 × 224 × 1), and $H^{*}$ and $L^{*}$ have pseudo RGB channel (224 × 224 × 3) as shown as high- and low-resolution in Figure 7. We correspondingly defined both high- and low-resolution network with 13 convolution layers, 5 max-pooling layers, and 3 fully connected layers as shown in Figure 7. A 3 × 3 filter is used in each convolution layer, ReLU is used for an activation function, and 2 × 2 max-pooling filters are used to maximize the receptive field [9]. Each fully connected layer has 4096 perceptrons, except for the last five layers. Finally, the soft-max operator returns the probability using returned five values. In this paper, the proposed multi-resolution network is combined to our pre-convolution layer described in Section 3.2.

### 3.4. Orthogonal Learning

To combine the results of low- and high-resolution networks, we define the average least square loss as
(11)$$L_{least}=\frac{1}{2B}\sum_{n=0}^{B}\sum_{i=1}^{N}{\left({g_{i}}^{n}-P_{i}\left(H_{n}^{*}\right)\right)}^{2}+{\left(g_{i}-P_{i}\left(L_{n}^{*}\right)\right)}^{2},$$
where $H_{n}$ and $L_{n}$ respectively represent the n-th high and low resolution image, *B* denotes the size of batch, and *N* is the vehicle type between C1 to C5. *g* represents the one-hot vector of size B×5, which is pre-labeled in our dataset described in Section 3.1. To reduce the correlation between two groups, we also define the orthogonal loss as [26]
(12)$$L_{o}=\frac{1}{B}\sum_{n=0}^{B}\sum_{i=1}^{2}\left(P_{i}\left(H_{n}^{*},L_{n}^{*}\right)P_{3}\left(H_{n}^{*},L_{n}^{*}\right)+P_{i}\left(H_{n}^{*},L_{n}^{*}\right)P_{4}\left(H_{n}^{*},L_{n}^{*}\right)++P_{i}\left(H_{n}^{*},L_{n}^{*}\right)P_{5}\left(H_{n}^{*},L_{n}^{*}\right)\right).$$

In binary classification, the error can be reduced when sum of multiplication between subcompact and other group is closed to zero as shown in Figure 8. Therefore, the proposed total loss can be defined as
(13)$$L_{total}=L_{least}+L_{o}.$$

To reduce the total loss $L_{total}$, the set of parameters including convolution kernel, bias, and perceptron weight are updated via stochastic gradient decent optimization [9]. The learning and dropout rates are set to 0.0001 and 0.5, respectively. For supervised learning, we train the model using 1500 labeled training data given in Section 3.1. For transfer learning, all of convolution layers are pretrained by ImageNet data [10]. We trained the proposed model using 4500 iterations, and 15 batches are engaged to the proposed multi-resolution network for each learning. Figure 9 shows the proposed learning procedure. Finally, to distinguish subcompact vehicle, the probability values of the optimized model are estimated using the thresholding operator as
(14)$$D_{n}=\left\{\begin{array}{cc} False & \arg\max\frac{1}{2}\left(P\left(H_{n}^{*}\right)+P\left(L_{n}^{*}\right)\right)\in \left\{C1,C2\right\}\\ True & \arg\max\frac{1}{2}\left(P\left(H_{n}^{*}\right)+P\left(L_{n}^{*}\right)\right)\in \left\{C3,C4,C5\right\} \end{array}\right.$$

## 4. Experimental Results

### 4.1. Quantitative Evaluation

To evaluate the proposed method, we compared experimental results with 2000 test data and state-of-the-arts classification models including HOG based recognition model (HOG+SVM) proposed by Dalal et al. [3], MCCF combined HOG recognition (MCCF+HOG+SVM) [15], multi-resolution image based HOG recognition (Retinex+MCCF+HOG+SVM), deep neural network based Huttunen’s method (DNN) [8], convolutional neural network (CNN) with 16 layers proposed by Simonyan et al. [9], retinex CNN Retinex+CNN, based on proposed pre-convolution layer (Retinex+MCCF+CNN), and the proposed multi-resolution network without orthogonal learning (Retinex+MCCF+MRN). All the algorithms were implemented in visual studio 2015 and Python 3.5 using on a desktop PC with i7 CPU, 64 GB RAM, and NVIDIA RTX 2080ti graphics processing unit (GPU). We also quantitatively measured accuracy (Acc.), precision, recall, and false-positive rate (FPR) as
(15)$$precision=\frac{TP}{TP+FP},$$
(16)$$Recall=\frac{TP}{TP+FN},$$
(17)$$FPR=\frac{FP}{TP+FP},$$
and
(18)$$Acc.=100\times \frac{TP+TN}{2000},$$
where TP, TN, FP, and FN respectively represent the true-positive, true-negative, false-positive, and false negative. Table 1 shows several evaluation results using state-of-the arts and the proposed methods. Since HOG uses handcraft-based features, its recognition performance is limited for ambiguous features. HOG+SVM method results in many mis-classification cases represented by FN and FP as shown in Table 1.

MCCF+HOG+SVM method can improve the false-positive case because MCCF based localization effectively removes unnecessary information such as background, but FN case can be increased. Since retinex-based image enhancement enhance too much textures, MCCF+HOG+SVM outperforms Retinex+MCCF+HOG+SVM in every sense. The deep neural network (DNN) can effectively increase the TP compared with SVM-based methods, but false-positive rate was slightly higher than MCCF+HOG+SVM. Simonyan’s convolutional neural network model can improve both TP and TN, so accuracy was highly increased over 4% than both DNN and SVM based methods, Especially, false-positive rate rapidly decreases compared with DNN and SVM based method, but accuracy is not enough because of the imbalance between true positive and false negative. Since the enhanced textures in a shadow region can compensate for the imbalance, the retinex-based CNN model (Retinex+CNN) outperforms the vanilla CNN in terms of the recall, accuracy. As a result of the localization error of MCCF, Retinex+MCCF+CNN can generate errors such as FN and FP, but the combined version Retinex+MCCF+MRN outperforms CNN models in all of evaluation terms. This is mean that the proposed MRN can adaptively reflect between localized information and enhanced textures to recognize the sub-compact vehicle. Furthermore, the proposed orthogonal learning method has TP values higher than Retinex+MCCF+MRN because it can generate the uncorrelated group-probability vectors. Note that the precision, recall, fpr, and accuracy are better by 0.1268, 0.45, 1.268, and 12.25% than convolutional neural network (CNN). In conclusion, the proposed approach can effectively classify the ambiguous objects because it designed and optimized with consideration of the group-error and ambiguous features. In addition, the multi-class recognition performance is compared with the CNN model as shown in Figure 10. The CNN method misclassifies between sedan and subcompact vehicles many times. Furthermore, subcompact truck and van are sometimes mis-recognized by the CNN method. However, the proposed method can not only reduce the mis-recognized case but also improve the accuracy by 10.85%. In addition, we conducted ablation study using validation check as shown in Figure 11. When we train the MRN without pseudo color (Equation 9), the performance can be degraded as shown black line in Figure 11 because the MRN was pre-trained by true color image dataset. The center crop-based MRN(Center crop) means that the MCCF operator in proposed pre-convolution layer was replaced to center cropping method [13]. Since the center cropping method can not adaptively localize the object, it can not outperform than proposed MRN. Furthermore, if MRN was optimized by only least square loss (Equation 11), the extracted features are not suitable for binary classification. Thus, when MRN was learned without proposed orthogonal loss (Equation 12), the performance of the MRN can not reach the orthogonal learning-based MRN as shown in Figure 11.

### 4.2. Baseline Comparison

To compare the efficient baseline networks, we evaluate the accuracy with several efficient networks such as VGG16 [9], residual network50(resnet50) [11], and resinext50 [12]. Table 2 shows the maximum binary, and multi class accuracy values. Since general networks do not consider ambiguous in binary classification problem, the MRN outperforms than several based line networks in terms of both binary and multi-class accuracy. The residual network based MRN slightly lower than the VGG16 based MRN because the VGG16 have already deep layers in binary classification. However, resnext based MRN outperforms the VGG16 based MRN in terms of binary accuracy because the split-transform-merge strategy can effectively apply to recognize the ambiguous binary objects. Figure 12 shows the accuracy for each training epoch. Note that the proposed networks outperform than state-of-the-arts baseline-networks in most epoch. In addition, we recorded computational complexity and allocated GPU-memory on average to verify the computational efficiency.

### 4.3. Visualization

To verify the performance of the proposed orthogonal learning, we visualized the output values of last layer in the both CNN [9] and the proposed MRN by projecting to two-dimensional space using t-stochastic neighbor embedding (t-SNE) [27]. In Figure 13a, the visualization is not easy because of the correlation between subcompact vehicle and other groups. However, Figure 13b shows that the points are clustered to easily classify, which means that the proposed orthogonal learning can remove the group-correlation. As a result, proposed orthogonal learning can improve the performance of deep binary classification.

In Figure 14, we visualized the classified label and localized regions, where the localized bonnet region is the input to the low-resolution network, and the entire image is engaged to the high-resolution network. The resulting label (subcompact and non-subcompact vehicle) based on the average value of the two probabilities is reflected at the end of red-box. The proposed MRN can not only classify in various illuminations but also distinguish the ambiguous vehicle types.

## 5. Conclusions

To recognize ambiguous features between the subcompact and other vehicles, we collected a novel set of subcompact vehicle images, and proposed a pre-convolution layer that is combined with the multi-resolution network with an orthogonal learning method. The proposed method can not only enhance the textures using retinex-based enhancement but also adaptively cropped the bonnet region using correlation computation. As a result, our MRN can avoid over-fitting by ambiguous features between vehicle types, and outperforms the existing CNN(VGG16) method by 12.25%. Therefore, the proposed method can be applied to various traffic management systems such as toll and parking gates for automatic charging system. In the future work, we will expand the proposed MRN by combining the license plate detection system. The source code is available at https://github.com/JoongcholShin.

## Figures and Tables

**Figure 1 sensors-20-02715-f001:**
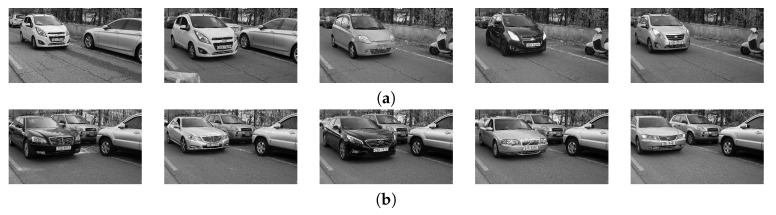
(**a**) Subcompact vehicles and (**b**) sedans. It is not easy to differentiate two classes using small features such as head lamp or rear-view mirror. On the other hand, there are differences in bigger features such as bonnet and overall shape of vehicles.

**Figure 2 sensors-20-02715-f002:**
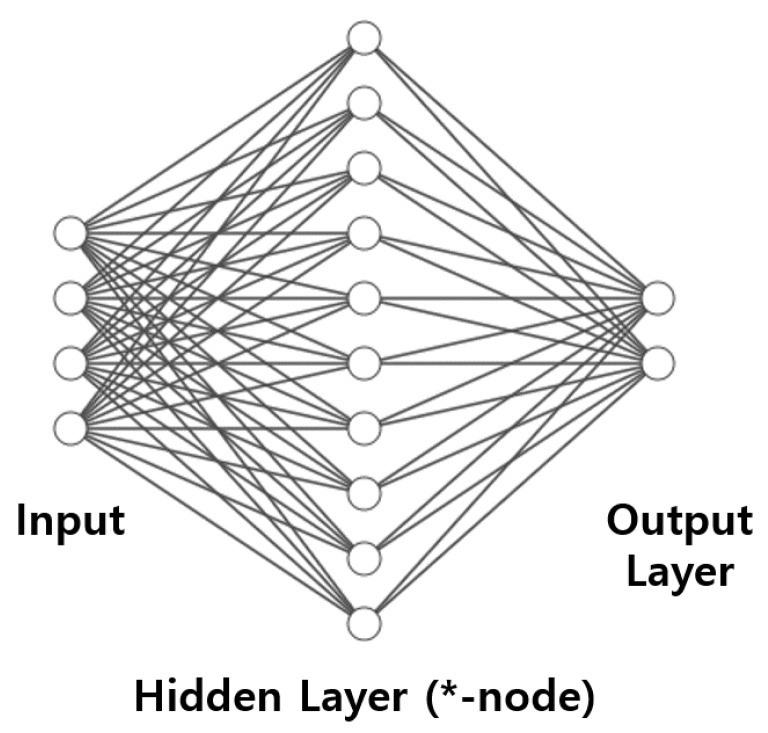
The architecture of neural network.

**Figure 3 sensors-20-02715-f003:**
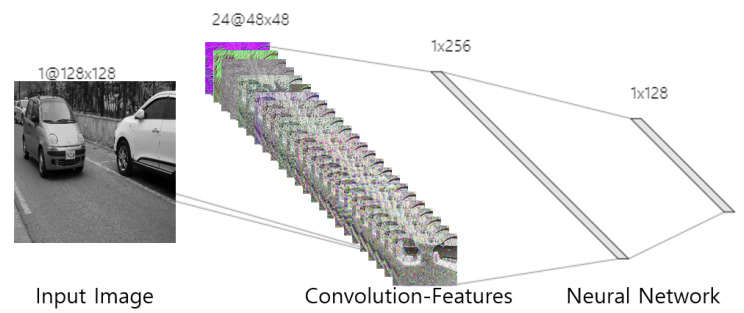
The convolutional neural network and convolution features.

**Figure 4 sensors-20-02715-f004:**
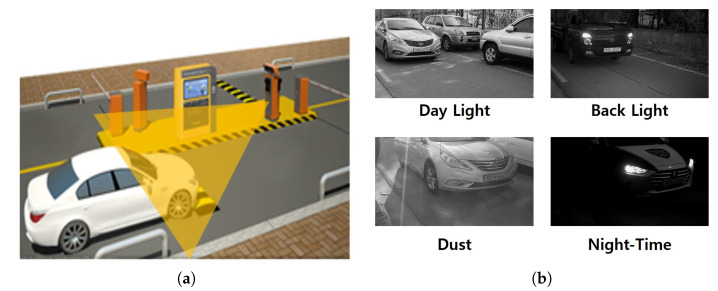
(**a**) Camera installation and (**b**) four different illumination conditions.

**Figure 5 sensors-20-02715-f005:**
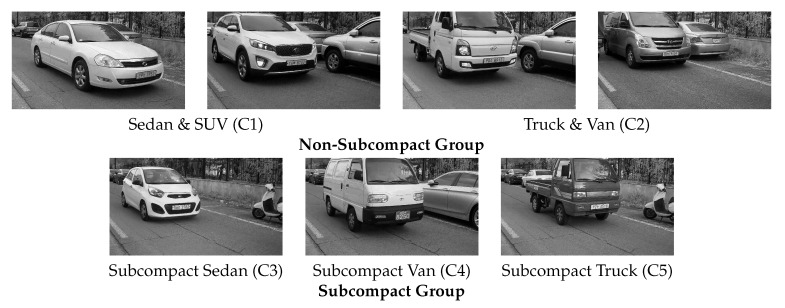
An example of five vehicle classes in the proposed dataset.

**Figure 6 sensors-20-02715-f006:**
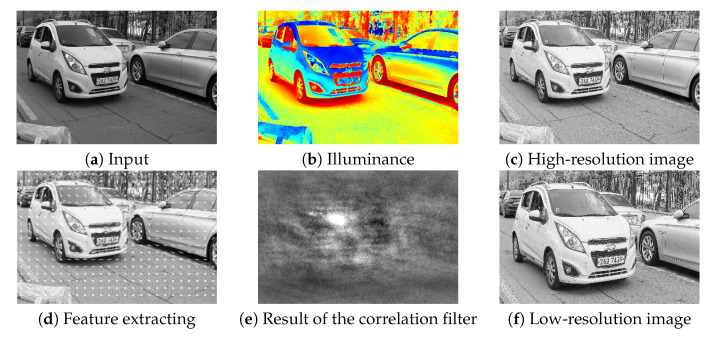
Step-by-step results in the pre-convolution layer (**a**–**f**).

**Figure 7 sensors-20-02715-f007:**
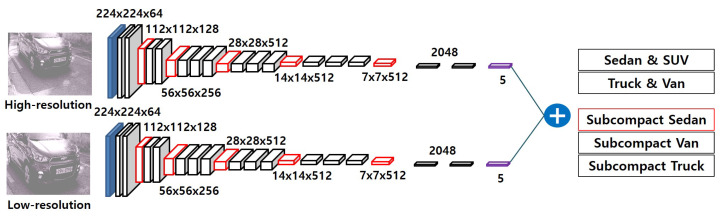
The architecture of the proposed multi-resolution network. Blue, black, and red cubes represent the resized input, convolution, and max-pool layers, respectively. Black and purple boxes are fully-connected and softmax layers, respectively.

**Figure 8 sensors-20-02715-f008:**
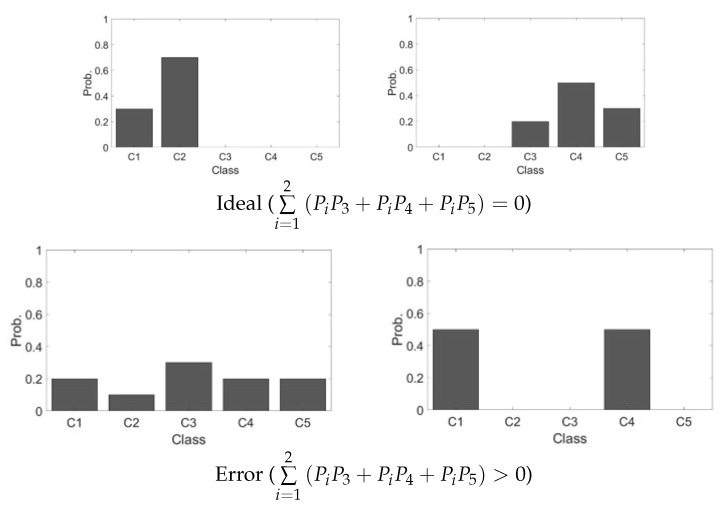
An example of orthogonal learning.

**Figure 9 sensors-20-02715-f009:**
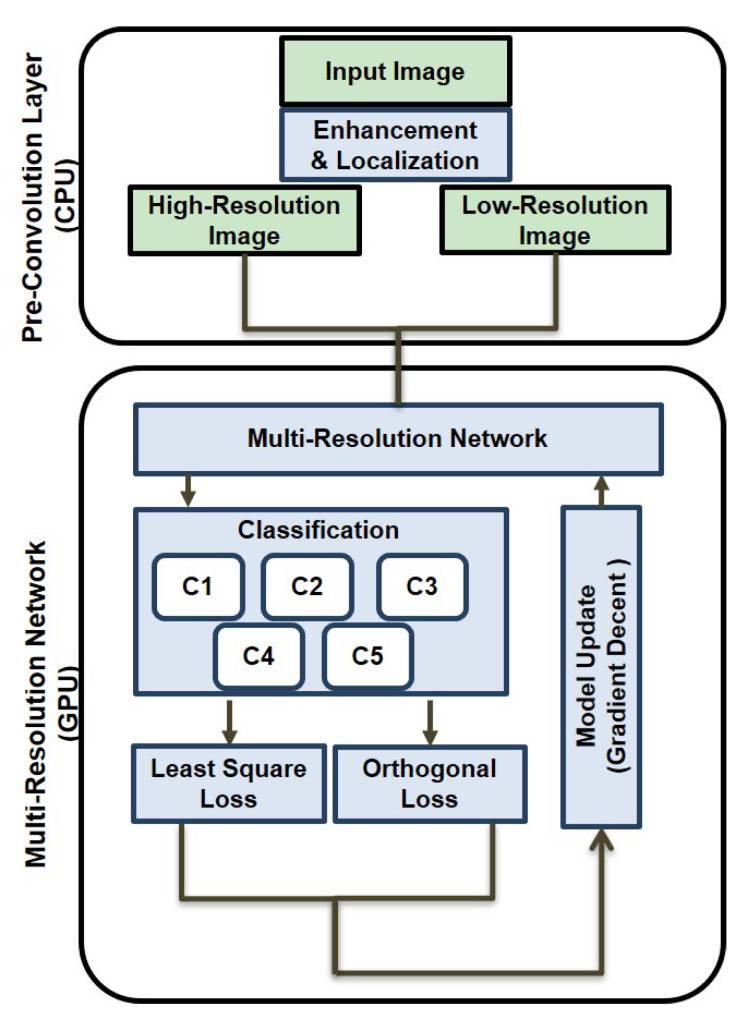
Dual procedure of the proposed method using both CPU and GPU.

**Figure 10 sensors-20-02715-f010:**
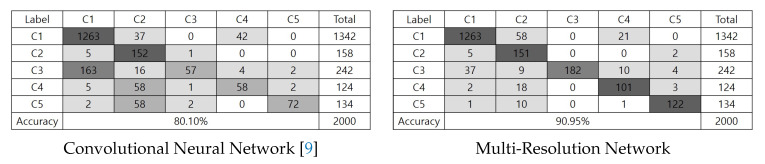
Multi-classification comparison.

**Figure 11 sensors-20-02715-f011:**
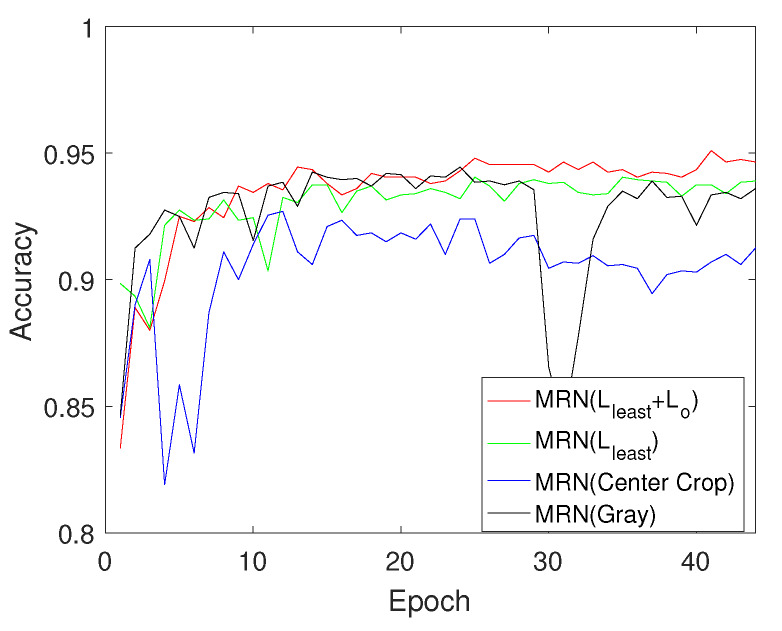
Ablation study: MRN ($L_{least}+L_{o}$) is the proposed orthogonal learning based multi-resolution network (MRN), MRN ($L_{least}$) is the least square loss based MRN, MRN (Center Crop) is the center cropping method to generate the low-resolution image, and MRN (Gray) is a single channel based MRN.

**Figure 12 sensors-20-02715-f012:**
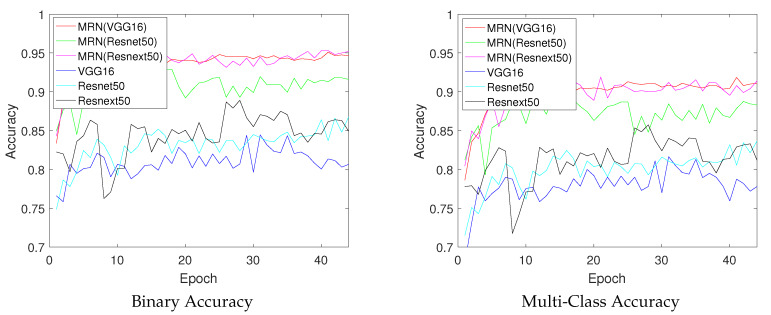
Accuracy evaluation according to each epoch.

**Figure 13 sensors-20-02715-f013:**
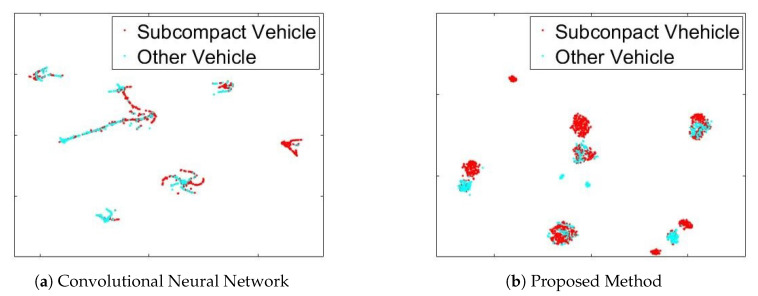
Probability Visualization using t-Stochastic Neighbor Embedding (t-SNE) [27].

**Figure 14 sensors-20-02715-f014:**
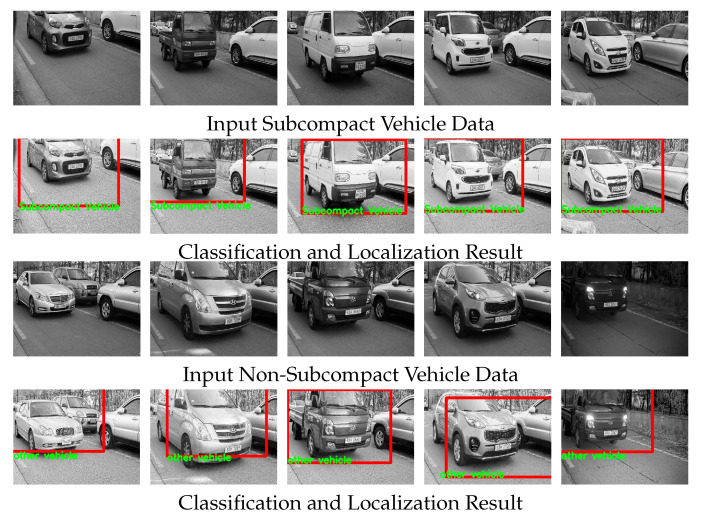
Classification and localization result.

**Table 1 sensors-20-02715-t001:** Quantitative comparison with state-of-the art approaches.

Method	TP	FN	TN	FP	Precision	Recall	FPR	Acc.
HOG+SVM	165	335	912	588	0.2191	0.3300	0.7809	53.85%
MCCF+HOG+SVM	43	457	1477	23	0.6515	0.0860	0.3485	76.00%
Retinex+MCCF+HOG+SVM	37	463	1455	45	0.4512	0.0740	0.5488	74.60%
DNN	182	318	1377	123	0.5967	0.3650	0.4033	77.95%
CNN	198	302	1457	43	0.8216	0.3960	0.1784	82.75%
Retinex+CNN	410	90	1474	26	0.9404	0.8200	0.0596	94.20%
Retinex+MCCF+CNN	373	127	1444	56	0.8695	0.7460	0.1305	90.85%
Retinex+MCCF+MRN	417	83	1478	22	0.9499	0.8340	0.0501	94.75%
ProposedMRN	423	77	1477	23	0.9484	0.8460	0.0516	95.00%

**Table 2 sensors-20-02715-t002:** Effects on the accuracy for different baseline networks.

Method	Baseline	Tool	Accuracy (Binary)	Accuracy (Multi)	Proc. Time (ms)	GPU-Memory (GB)
CNN	VGG16	Tensorflow	0.8275	0.8010	65 ms	1.3 GB
CNN	Resnet50	Pytorch	0.8740	0.8435	80 ms	1.0 GB
CNN	Resnext50	Pytorch	0.8890	0.8575	86 ms	1.0 GB
MRN	VGG16	Tensorflow	0.9500	0.9095	70 ms	1.6 GB
MRN	Resnet50	Pytorch	0.9290	0.8810	100 ms	1.2 GB
MRN	Resnext50	Pytorch	0.9530	0.8955	106 ms	1.3 GB

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
