# Peer review of "Deep Binary Classification via Multi-Resolution Network and Stochastic Orthogonality for Subcompact Vehicle Recognition"

_sensors, 2020, doi:10.3390/s20092715_

Round 1

Reviewer 1 Report

This paper is about subcompact vehicle recognition.
Their proposed method is based on combination of
Retinex, MCCF, multi resolution network, orthogonal optimization.

The evaluation is systematically done, and authors confirmed the effectiveness of the proposed method.

There are a few points that are ambiguous, so please add an explanation for them.

About dataset details:
I guess the proposed method is strongly restricted to the specific situation.
Could you please explain about the details of the dataset.
The readers need variation of the dataset.
- Resolution and number of channels(RGB or Gray?)
- At where were these images taken? One place, two places? Does this dataset have angular variation?
- At what time were these images taken? Does this dataset include temporal variation? (Different illumination condition?)
- Is there any seasonal variation in this data set? 3500 images for the specific day or some days or one month, one year? (Different background?)

Figure 1 appears to be squared off and the aspect ratio has been changed.
Here, authors need to show the original images from the dataset.

Figure 2, the input image is gray scale, but 3 channels.
It is confusing. The author must use color image if the input has 3 channels.
It is the same for Fig.4.

In section3.2:
About eq.(5) and eq.(6), is this enhancement process channel-wise, or gray scale, or something else?
About eq.(7),
-What are i,j,N and D? And how much is K in this study?
-Could you explain about the meaning of "x_i(\Delta\tau_j) is a set of cyclically shifted vehicle images in the training dataset"?
-Why y_i and x_i are cyclic?
-I understand this equation to find w which gives the smallest E(w).
Then the target of this equation y_i is important.
* What is response map y_i? There is no explanation about this variable.
* Why this variable is Gaussian-shaped?
* How do authors determine y_i in this study?

l.91 About "We cropped this coordinates to 92 obtain low-resolution image L."
The proposed point is multi-resolution. Therefore this is important point.
I feel lack of explanation.
How authors determine the cropped region? Just find peak of the response map and set it to the center of fixed size rectangle(224x224)?
Please explain more details.

l.92
"Fig.5 shows the block diagram." seems to be a mistake.
Figure 5 is not block diagram. It shows some images.
(The sentence just under eq(6) also cite figure 5. The author may have forgotten Figure 6)

In section3.3
How do you make high resolution images for the network inf Fig6?
Just resize from original images with changing aspect racio?
How about low resolution image? Is it cropped after aspect ratio change, or before aspect ratio change?

In section4.
"To evaluate the proposed method, we compared experimental results with 2000 test data"
How did you determine the model of neural networks(DNN,CNN,proposed) to evaluate test data?
Basically, we divide dataset into 3, train/validation/test, and apply the model which shows the best in validation to test data.
In this study, are the best results for test data of each models shown?

Author Response

We attached the list of changes for the reviewers' comments of the submitted paper “sensors-761477”.
Please check the attached files.

Reviewer 2 Report

This paper studies a binary classification problem for recognizing subcompact cars.The proposed method firstly employs retinex-based image enhancement and multi-channel correlation filter to get high-resolution region image. The extracted high-resolution region image and the original low-resolution image are then put into two parallel networks and finally the outputs are fused at classification level to give the final results. Here are some comments on the paper:

1. The design and training methods of the model used in this paper are too old, probably staying at the level of 2014. In recent years, the research on image classification algorithms has made rapid progress, and a large number of research results with superior performance, higher efficiency and wide application have emerged, such as ResNet(CVPR2016), DenseNet(CVPR2017),  PeleeNet(NIPS2018), ThunderNet(ICCV2019), and EfficientNet(ICML2019). It is suggested to pay more attention to the current situation of technology development.

2.In the experimental verification, the contribution of each design module to the algorithm should be fully verified.In addition, in order to verify the superiority of the design module, it is necessary to make a comparative analysis with the existing mainstream modules. The experimental logic in this paper is not clear enough, and some content is missing.

3.There are some obvious grammatical errors in the paper.

Author Response

(The authors gave the same response as above.)

Reviewer 3 Report

To recognize ambiguous features between the subcompact and other vehicles, the authors collected a novel set of subcompact vehicle images, and proposed a pre-convolution layer that is combined with the multi-resolution network with an orthogonal learning method. The proposed method can not only enhance the textures using retinex-based enhancement but also adaptively cropped the bonnet region using correlation computation. There are some questions for the authors to answer:

1) The method used in Pre-Convolution Layer is not very new, image enhancement is a retinex-based algorithm. The MCCF method to obtain low-resolution wouldn’t be a good choice. There are more new methods to get a low-resolution image.

2) Retinex + CNN is much better than Retinex + MCCF + CNN, the effectiveness of MCCF seems to be in doubt, and Retinex + MRN should be added for comparison (generated low-resolution image using other simple methods).

3) The performance gap between Retinex + MCCF + MRN and the proposed MRN is not very large, and the validity of orthogonal loss is not reflected. The difference may be due to bias in training experiments.

4) The test dataset of this method is not rich enough, only on the self-built dataset to complete the relevant experiments, there is no generalization comparison.

5) The input image requirements are relatively fixed, which can only work in one scene, the camera, the shooting vehicle distance and angle are fixed.

6) References seem to be older, less in the last few years.

Author Response

(The authors gave the same response as above.)

Round 2

Reviewer 1 Report

Thank you for revising your article.
I understand the method.
I think pseudo-color is no-meaning(doesn't carry any information), but I guess you may need this part to direct comparison between other method.
And Fig6(e) shows the peak of the correlation filter is not on the bonnet region.(It is on the side of the car).
It is not I expected according to your article, but you may choose a low resolution area by shifting the peak instead of centering it.

Author Response

(The authors gave the same response as above.)

Reviewer 2 Report

In general, the paper uses a lot of processing skills, but not fully demonstrated. Although a little verification has been added to the revised draft, there are still some problems, such as the lack of emphasis and innovation. Here are some important examples:

1. In the preprocessing stage, the authors use MCCH to extract local image blocks (280 * 200), which is a method ten years ago. Now the mainstream algorithm uses fully convolutional network, which can achieve the same form of dense response output as correlation filtering better and faster. This is not verified in the paper.

2. The authors design a feature extraction network based on experience (Figure 7), but it is not widely compared with the existing model in terms of performance and efficiency. The verification added in the revised version (Table 2) only shows the effectiveness of the two branch strategy, and cannot be used to verify the effectiveness of the proposed feature extraction network.

3. As a whole, the paper is written as a technical report. The implementation steps are listed in order, and the refinement and verification of innovation are ignored. In addition, as an application-oriented work, algorithm efficiency is also very important. The paper lacks time-consuming analysis of each module. There is no doubt that the two branch strategy has greatly increased the computation, and the efficiency comparison between it and the single branch method is equally important for practical application.

Author Response

(The authors gave the same response as above.)

Reviewer 3 Report

The paper have answered some of my doubts and can be accepted.